# Curriculum-guided Hindsight Experience Replay

**Meng Fang**[1]*, **Tianyi Zhou**[2]*, **Yali Du**[3], **Lei Han**[1], **Zhengyou Zhang**[1]

[1]Tencent Robotics X
[2]Paul G. Allen School of Computer Science & Engineering, University of Washington
[3]University College London

## Abstract

In off-policy deep reinforcement learning, it is usually hard to collect sufficient successful experiences with sparse rewards to learn from. Hindsight experience replay (HER) enables an agent to learn from failures by treating the achieved state of a failed experience as a pseudo goal. However, not all the failed experiences are equally useful to different learning stages, so it is not efficient to replay all of them or uniform samples of them. In this paper, we propose to 1) adaptively select the failed experiences for replay according to the proximity to true goals and the curiosity of exploration over diverse pseudo goals, and 2) gradually change the proportion of the goal-proximity and the diversity-based curiosity in the selection criteria: we adopt a human-like learning strategy that enforces more curiosity in earlier stages and changes to larger goal-proximity later. This "Goal-and-Curiosity-driven Curriculum Learning" leads to "Curriculum-guided HER (CHER)", which adaptively and dynamically controls the exploration-exploitation trade-off during the learning process via hindsight experience selection. We show that CHER improves the state of the art in challenging robotics environments.

## 1 Introduction

Deep reinforcement learning (RL) has been an effective framework addressing a rich repertoire of complex control problems. In simulated domains, deep RL can train agents to perform a diverse array of challenging tasks [Mnih et al., 2015, Lillicrap et al., 2015, Duan et al., 2016]. In order to train reliable agents, it is critical to not only design a reward faithfully reflecting how successful the task is, but also (re)shape the reward [Ng et al., 1999] to provide dense feedback that can efficiently guide the policy optimization towards a better solution in the given environment. Unfortunately, many of the capabilities demonstrated by the current reward engineering are often limited to specific tasks in specified environments. Moreover, the quality of reward shaping heavily relies on both the choice of RL algorithm and domain-specific knowledge. For situations where we do not know what admissible behavior may look like, for example, using LEGO bricks to build a desired architecture, it is difficult to apply reward engineering. Therefore, it is essential (but also challenging) to develop smarter and more general algorithms which can directly learn from unshaped and usually sparse reward signals, where the sparsity is caused by the insufficiency of successful experiences (which are expensive to collect). Hindsight Experience Replay (HER) [Andrychowicz et al., 2017] proposes to additionally leverage the rich repository of the failed experiences, by replacing the desired (true) goals of training trajectories with the achieved goals of the failed experiences. With this modification, any failed experience can have a nonnegative reward.

The achieved goals of failed experiences can be significantly different to each other: their proximity to the desired goal varies so learning how to reach a pseudo goal distant from the true one cannot directly help the targeted task; they also carry different information about the manipulation environment.

Hence, they have distinct levels of difficulty to be learned and their contributions to a task vary across different learning stages. Nevertheless, they are treated equally in HER: they are uniformly sampled to replace the desired goals; and the resulted training trajectories with the replaced goals have the same weight in training. However, not all the failed experiences are equally useful to improve the agent: some provide limited helps to reach the true goal; while some are too similar to each other and thus redundant to be all learned.

In human education, a delicately designed curriculum can significantly improve the learning quality and efficiency. Inspired by this, curriculum learning [Bengio et al., 2009] and its applications [Khan et al., 2011, Basu and Christensen, 2013, Spitkovsky et al., 2009] propose to train a model on a designed sequence of training samples/tasks, i.e., a curriculum, which leads to improvement on both learning performance and efficiency. In each learning stage, the training samples are selected either by a human expert or an adaptive algorithm, and the selection can be either pre-defined before training begins or determined by the learning progress itself on the fly [Kumar et al., 2010]. Most curriculum learning methods, e.g., self-paced learning and its variants [Tang et al., 2012a, Supancic III and Ramanan, 2013, Tang et al., 2012b], adopt the strategy of selecting a few easier training samples at beginning and an increased amount of more difficult ones later on. Recent works [Zhou and Bilmes, 2018, Zhou et al., 2018] show that diversity also needs to be considered in curriculum generation. Curriculum learning has been explained as a form of continuation scheme [Allgower and Georg, 2003] that addresses a hard task by solving a sequence of tasks moving from easy to hard, and uses the solution to each task as the warm start for the next slightly harder task. Such continuation schemes can reduce the impact of local minima within neural networks [Bengio et al., 2013, Bengio, 2014].

In this paper, we propose "Goal-and-Curiosity-driven Curriculum Learning" that dynamically and adaptively controls the exploration-exploitation trade-off in selecting hindsight experiences for replay by gradually changing the preference on 1) goal-proximity: how close the achieved goals are to the desired goals; and 2) diversity-based curiosity: how diverse the achieved goals are in the environment. Specifically, given a candidate subset of achieved goals for HER training, we define its proximity as the sum of their similarities to the desired goals, and measure its diversity by a submodular function [Fujishige, 2005], e.g., the facility location function [Cornuéjols et al., 1977, Lin et al., 2009]. In each episode, a subset of achieved goals are selected according to both its proximity and curiosity: we prefer more curiosity for earlier episodes' exploration and then gradually increase the proportion of proximity in the selection criteria during later episodes. We apply this training framework, called "Curriculum-guided HER (CHER)", to train agents in the multi-goal setting of UVFA [Schaul et al., 2015] and HER [Andrychowicz et al., 2017] (which assumes that the goal being pursued does not influence the environment dynamics). In several challenging robotics environments (where deep RL methods suffer from sparse reward problem), CHER outperforms the state-of-the-art approaches on both the learning efficiency and final performance.[1]

## 2 Related Work

In recent works, curriculum learning with progressive training strategy has been introduced to different scenarios of deep RL. Those methods differ in that they apply an increasing difficulty/complexity scheduling to different components of the training loop, e.g., the initial positions [Florensa et al., 2017], the required $\epsilon$-accuracy [Fournier et al., 2018], the policies of intermediate agents used for mixing [Czarnecki et al., 2018], the environments [Wu and Tian, 2017], the aid from built-in AI [Tian et al., 2017], the reward [Justesen and Risi, 2018], and new tasks with masked sub-goals [Eppe et al., 2018]. These works show that curriculum learning can effectively improve deep RL for challenging tasks including robotics manipulation, game-bot, and simulated environment such as OpenAI Gym. HER can also be explained as a form of implicit curriculum learning, since the achieved goals of failed experiences are easier to achieve than the desired goals. The last work mentioned above improves HER but requires extra efforts in each epoch to evaluate the difficulty of sub-goals and train the new tasks with sub-goals. It is not practical for tasks with complex goals, such as the hand manipulation tasks studied in this paper.

Another line of recent work [Burda et al., 2019, Pathak et al., 2017, Savinov et al., 2019, Frank et al., 2013] investigates the curiosity-driven exploration of deep RL agents within interactive environments. In particular, they either replace or augment the extrinsic (but usually sparse) reward by a dense

intrinsic reward measuring the curiosity or uncertainty of the agent at a given state. Thereby, the agent is encouraged to explore unseen scenarios and unexplored regions of the environment. It has been shown that such curiosity driven strategy can improve the learning efficiency, mitigate the sparse reward problem, and successfully learn challenging tasks even without extrinsic reward. Different from curriculum learning approaches, which are usually goal-oriented with focus on exploitation, curiosity-driven approaches can be unsupervised/self-supervised with more focus on exploration. Comparing to our strategy, they reshape the reward but do not dynamically and adaptively change the proportion of curiosity in the reward during training.

A number of RL methods leveraging hindsight experiences have been proposed since HER. Hindsight Policy Gradient (HPG) [Rauber et al., 2019] extends the idea of training goal-conditional agents on hindsight experiences to on-policy RL setting. Dynamic Hindsight Experience Replay (DHER) [Fang et al., 2019] assembles failed experiences to train policies handling dynamic goals rather than static ones studied in HER. On top of HER, Competitive Experience Replay (CER) [Liu et al., 2019] introduces a competition between two agents for better exploration. To handle raw-pixel inputs, Nair et al. [2018] minimize a pixel-MSE given visual observations with an extra cost of training a VAE. Zhao and Tresp [2018] focus on hindsight trajectories containing higher energy than others and claim that they are more valuable to training. Unlike the above works, our curriculum learning scheme can be generalized to a variety of settings and environments for more efficient goal-conditional RL.

## 3 Methodology

In this section, we will briefly introduce HER and multi-goal RL at first. Then, we will study the selection criteria applied to hindsight experiences. An efficient selection algorithm is introduced afterwards. In the end, we will present CHER with scheduled goal-proximity and diversity-based curiosity in the selection criteria.

### 3.1 HER and Multi-Goal RL

We study an agent operating in a multi-goal environment with sparse reward [Schaul et al., 2015, Andrychowicz et al., 2017]. At each time step $t$, the agent gets an observation (or state) $s_t$ from the environment and takes an action $a_t$ in response by applying its policy $\pi(s_t)$ (deterministic policy maps $s_t$ to $a_t = \pi(s_t)$, while stochastic policy samples $a_t \sim p(a_t|s_t) = \pi(s_t)$), then it receives a reward signal $r_t = r(s_t, a_t)$ and gets the next state $s_{t+1}$ sampled from the transition probability $p(\cdot|s_t, a_t)$. Given a behavior policy $\pi(\cdot)$, the agent can generate a trajectory $\tau = \{(s_0, a_0), \cdots, (s_{T-1}, a_{T-1})\}$ of any length $T$, with each step $t$ associated with a transition tuple $(s_t, a_t, r_t, s_{t+1})$. In many RL tasks, the reward only depends on whether the trajectory finally reaches a desired goal $g$ or not. Hence, only the successful trajectories get nonnegative rewards. Since $\pi(\cdot)$ is not fully-trained and has low success rate, the collected successful trajectories are usually insufficient for training, which results in the sparse reward problem.

HER addresses the sparse reward problem by treating failures as successes and learning from the failed experiences. For any off-policy RL algorithm (e.g., DQN [Mnih et al., 2015], DDPG [Lillicrap et al., 2015], NAF [Gu et al., 2016], SDQN [Metz et al., 2017]), HER modifies the desired goals $g$ in the transition tuples for training to some achieved goals $g'$ sampled from the states in failed experiences. The desired goal $g$ is the actual goal that the agent aims to achieve, i.e., the real target. An achieved goal $g'$ is a state that the agent has already achieved, e.g., the Cartesian position of each fingertip on a robotic hand. Once $g$ is replaced by a $g'$, the corresponding failed experience is assigned a nonnegative reward and thus can contribute to learning policies.

### 3.2 Goal-and-Curiosity-driven Selection of Pseudo Goals

In HER, the achieved goals used to modify the desired goals are uniformly sampled from (a batch of) previous experiences $B$. In contrast to uniform sampling, we propose to select a subset of achieved goals $A \subseteq B$ according to 1) their proximity to the desired goals and 2) their diversity that reflects the curiosity of an agent exploring different achieved goals in the environment. Although all the failed experiences can be turned into success ones with pseudo goals, they can be very different in the above two quantities, which however play important roles in guiding the learning process. In particular, a large proximity enforces the training to proceed towards the desired goals, while a large diversity

leads to more exploration of different states and regions in the environment. A desirable trade-off between them is essential to the learning efficiency and generalization performance of resulted agents.

In our selection criteria, we select a subset of failed experiences to replay according to its proximity and diversity, which are both defined based on a similarity function $\text{sim}(\cdot, \cdot)$ measuring the likeness of two achieved goals in the interactive environment. Given a distance metric $\text{dis}(\cdot, \cdot)$ (e.g., Euclidean distance), $\text{sim}(\cdot, \cdot)$ can be defined, for example, by the radial basis function (RBF) kernel with bandwidth $\sigma$, i.e.,

$$\text{sim}(x, y) \triangleq \exp\left(\frac{-\text{dis}(x, y)^2}{\sigma^2}\right), \tag{1}$$

while another option is a constant $c$ minus the distance, i.e.,

$$\text{sim}(x, y) \triangleq c - \text{dis}(x, y), \tag{2}$$

where $c$ is large enough to guarantee that $\text{sim}(x, y) \geq 0$ for all possible $(x, y)$. The choice of $\text{dis}(\cdot, \cdot)$ is usually determined by the task and environment. For instance, in hand manipulation tasks, we can define $\text{dis}(g_i, g_j)$ as the mean distance between fingertips at time step $i$ and fingertips at time step $j$.

We are now able to select a subset $A$ of achieved goals with size up to $k$ from buffered experiences $B$ by solving the following combinatorial optimization that maximizes both the proximity and diversity:

$$\max_{A \subseteq B, |A| \leq k} F(A) \triangleq \lambda F_{prox}(A) + F_{div}(A). \tag{3}$$

The first term $F_{prox}(A)$, associated with a trade-off weight $\lambda$, is a modular function

$$F_{prox}(A) \triangleq \sum_{i \in A} \text{sim}(g_i, g), \tag{4}$$

which reflects the proximity of the selected achieved goals $g'$ in $A$ to its desired goal $g$. The second term $F_{div}(A)$ measures the diversity of the goals from $A$. We use the facility location function [Cornuéjols et al., 1977, Lin et al., 2009] to compute $F_{div}(A)$, i.e.,

$$F_{div}(A) \triangleq \sum_{j \in B} \max_{i \in A} \text{sim}(g_i, g_j). \tag{5}$$

Intuitively, we expect the achieved goals selected into $A$ can represent all the goals from $B$. For each $g_j$ from $B$, $F_{div}(A)$ finds an achieved goal $g_i$ most similar to $g_j$ from $A$, and uses $\text{sim}(g_i, g_j)$ to measure how well $A$ can represent $g_j$. Hence, by summing up $\text{sim}(g_i, g_j)$ over all the achieved goals $j \in B$, $F_{div}(A)$ quantifies how representative of $A$ w.r.t $B$. It has been widely used as a diversity metric, because a large $F_{div}(A)$ indicates that every goal in $B$ can find a sufficiently similar goal in $A$, in other words, $A$ spans the space of $B$. A diverse subset $A$ of achieved goals encourage the agent to explore new states and unseen areas of the environment and thus learn to reach different goals.

The facility location function is a typical example from a large expressive family of submodular functions that satisfies the diminishing return property: given a finite ground set $V$, any $A \subseteq B \subseteq V$ and an element $v \notin B, v \in V$, they fulfill $F(\{v\} \cup A) - F(A) \geq F(\{v\} \cup B) - F(B)$ (with abuse of former notations $A$ and $B$). Due to this property, they can naturally measure the diversity of a set of items [Fujishige, 2005], and has been applied in a variety of diversity-driven tasks achieving appealing results [Lin and Bilmes, 2011, Batra et al., 2012, Prasad et al., 2014, Gillenwater et al., 2012, Fiterau and Dubrawski, 2012]. Although we choose facility location function to be $F_{div}(A)$ in this paper, other submodular functions are worth studying in our curriculum learning framework.

Since $F(A)$ in Eq. (3) is a weighted sum of a non-negative (the similarity is non-negative) modular function $F_{prox}(A)$ and a submodular function $F_{div}(A)$, it is monotone non-decreasing submodular. Although exactly solving Eq. (3) is NP-hard, a near-optimal solution can be achieved by the greedy algorithm with a worst-case approximation factor $\alpha = 1 - e^{-1}$ [Nemhauser et al., 1978], as a result of the submodularity of $F(A)$. The greedy algorithm starts with $A \leftarrow \emptyset$, and selects the next goal $i \in B \backslash A$ bringing the largest improvement $F(i|A) \triangleq F(i \cup A) - F(A)$ to the objective $F(A)$, i.e., $A \leftarrow A \cup \{i^*\}$ where $i^* \in \arg\max_{i \in B \backslash A} F(i|A)$, and this repeats until $|A| = k$. For the specific $F(A)$ defined in Eq. (3)-Eq. (5),

$$F(i|A) = \lambda \, \text{sim}(g_i, g) + \sum_{j \in B} \max\left\{0, \text{sim}(g_i, g_j) - \max_{l \in A} \text{sim}(g_l, g_j)\right\}. \tag{6}$$

**Algorithm 1** STOCHASTIC-GREEDY($k, m, \lambda$)

**Require:** experience buffer $\mathcal{B}$
1: **Input:** $k, m, \lambda$
2: **Output:** a minibatch $A$ of size $k$
3: Sample a batch $B$ of size $\mathcal{O}(mk)$ from $\mathcal{B}$, and build a sparse $K$-nearest neighbor graph of $B$.
4: Initialize $A \leftarrow \emptyset$;
5: **for** $i = 0$ **to** $k - 1$ **do**
6:     Sample a subset $b$ of size $m$ from $B \backslash A$;
7:     **for** each transition tuple $(s_t, a_t, r_t, s_{t+1})$ in $b$ **do**
8:         Calculate the utility score $F(i|A)$ by Eq. (6) (using $g_i = g'_t$ and $g$) based on the current state $s_t$;
9:     **end for**
10:    Add to $A$ the transition tuple that has the maximum utility score $F(i|A)$;
11: **end for**

The evaluation of $F_{div}(A)$ and $F(i|A)$ requires the pairwise similarity of any two goals $(g_i, g_j)$, and can be expensive when the size of $B$ is large. In practice, we can use kd-tree or ball-tree to build a sparse $K$-nearest neighbor graph for the goals in $B$ before running the greedy algorithm. It has been shown in previous works [Wei et al., 2014] that a sufficiently good solution can be achieved even for $K$ as small as $\mathcal{O}(\log |B|)$.

## 3.3 Lazier than Lazy Greedy for Efficient Selection

The greedy algorithm is simple to implement and usually outperforms other optimization methods, e.g., those based on integer linear programming, but suffers from expensive computation requiring $\mathcal{O}(|B|k)$ function evaluations. There exist several accelerations, e.g., lazy greedy [Minoux, 1978], lazier than lazy greedy [Mirzasoleiman et al., 2015] and GreeDi [Mirzasoleiman et al., 2016], which either has the same or close approximation factor but enjoys significant speedups.

We choose lazier than lazy greedy for the speedup of selecting failed experiences in CHER, because it is compatible with the stochastic learning nature of most off-policy RL algorithms. Algorithm 1 shows the detailed procedures. In each step, from a random subset $b$ of $B \backslash A$ (instead of $B \backslash A$) it selects the goal that results in the largest improvement $F(i|A)$. According to [Mirzasoleiman et al., 2015], when $m = \mathcal{O}(|B|/k \log 1/\epsilon)$, lazier than lazy greedy reduces the approximation factor $\alpha$ of the vanilla greedy algorithm by $\epsilon$, but only requires $\mathcal{O}(|B| \log 1/\epsilon))$ evaluations of $F(\cdot)$.

## 3.4 Curriculum-guided Hindsight Experience Replay

The trade-off between proximity and diversity in the selection of achieved goals reflects the trade-off between exploitation and exploration. Similar to the learning process of human, which requires different proportions of exploitation and exploration in different learning stages, the preference of proximity and diversity (or curiosity) in different episodes of deep RL also needs to vary. In the earlier episodes, curiosity over diverse pseudo goals can help the agent to explore new states and unseen areas. Thus it evolves RL for better generalization. However, diverse goals can distract the learning of later episodes, in which the proximity to the desired goals is more important for the agent since it has already accumulated sufficient knowledge about an environment and needs to focus on learning how to achieve the true goals of a task. Another critical reason to avoid large proximity in earlier episodes but promote it later is: the agent policy in earlier episodes cannot produce sufficient number of pseudo goals close to the desired goals (otherwise the learning is almost accomplished and we never suffer from sparse reward), but after adequate training it is able to do so later.

In the following, we propose "Goal-and-Curiosity-driven Curriculum (GCC) Learning" as an effective learning scheme for CHER. It starts from learning to reach different achieved goals with large diversity, and gradually turns the focus on how to progressively approach the achieved goals with large proximity to the desired goals. This is achieved by smoothly increasing the weight $\lambda$ of the proximity term in $F(A)$ of Eq. (3). For the tasks in this paper, we use an exponentially increasing $\lambda$ over the course of training, i.e.,

$$\lambda = (1 + \eta)^{\gamma} \lambda_0, \tag{7}$$

**Algorithm 2** Curriculum-guided HER (CHER)

---

**Require:** off-policy RL algorithm $\mathbb{A}$, experience buffer $\mathcal{B}$
 1: **Input:** mini-batch size $k$, $m$, $\lambda_0$, reward function $r(\cdot)$
 2: Initialize $\mathbb{A}$, $\mathcal{B} \leftarrow \emptyset$, $\lambda \leftarrow \lambda_0$;
 3: **for** episode $= 0, 1, \cdots, M-1$ **do**
 4:    Sample an initial goal $g$ and an initial state $s_0$
 5:    **for** $t = 0, \cdots, T-1$ **do**
 6:       Sample an action $a_t$ from the behavioral policy of $\mathbb{A}$, i.e., $a_t \sim \pi(s_t|g)$;
 7:       Execute action $a_t$ and observe a new state $s_{t+1}$;
 8:    **end for**
 9:    **for** $t = 0, \cdots, T-1$ **do**
10:       $r_t := r(s_t, a_t, g)$;
11:       Store the tuple $(s_t|g, a_t, r_t, s_{t+1}|g)$ in $\mathcal{B}$;
12:    **end for**
13:    **for** $i = 0, 1, \cdots, N-1$ **do**
14:       Select a subset $A$ of the achieved goals of $\mathcal{B}$ by Algorithm 1, i.e.,
           $A \leftarrow$ STOCHASTIC-GREEDY$(k, m, \lambda)$;
15:       Initialize a minibatch $B_i \leftarrow \emptyset$;
16:       **for** $g' \in A$ **do**
17:          $r' := r(s_t, a_t, g')$, where $\exists (s_t, a_t) \in \tau$: $g'$ has been achieved by $\tau$ after $t$;
18:          Store the tuple $(s_t|g', a_t, r', s_{t+1}|g')$ in $B_i$;
19:       **end for**
20:       Optimize $\mathbb{A}$ using minibatch $B_i$;
21:    **end for**
22:    $\lambda \leftarrow (1+\eta)\lambda$;
23: **end for**

---

where $\eta \in [0, 1]$ is a learning pace controlling the progress of the curriculum, $\gamma$ is the episode of the off-policy RL, and $\lambda_0$ is the initial weight for proximity, which should be relatively small.

The complete procedures of "Curriculum-guided HER (CHER)" can be found in Algorithm 2. Comparing to the vanilla HER, the major differences are at line-14, which selects the achieved goals from the experience buffer according to proximity and diversity, and line-22, which increases the weight for proximity as instructed by the curriculum. The algorithm can be generalized to improve any off-policy RL method, and does not require any extra training on new tasks.

Although Algorithm 1 cannot exactly solve the combinatorial optimization in Eq. (3), it is worth noting that the approximate solution can gradually approach the global optimal as the curriculum proceeds and $\lambda$ increases. Increasing $\lambda$ makes $F(A)$ close to a modular function. As a result, the greedy solution approaches the top-$k$ ranking, which is the optimal solution to modular maximization. This trend can be theoretically analyzed by the curvature-dependent approximation bound of greedy algorithm (which can easily extend to lazier than lazy greedy). It improves $\alpha = 1 - e^{-1}$ to $\alpha = (1 - e^{-\kappa_F})/\kappa_F$ [Conforti and Cornuejols, 1984], where the curvature $\kappa_F \in [0, 1]$ of $F(A)$ is defined as

$$\kappa_F \triangleq 1 - \min_{j \in B} \frac{F(j|B \backslash j)}{F(j)}. \tag{8}$$

When $\kappa_F = 0$, $F(\cdot)$ is modular, resulting in $\alpha = 1$ (which achieves the global optimum); and when $\kappa_F = 1$, $F(A)$ is fully curved and $\alpha = 1 - e^{-1}$. In CHER, when $\lambda$ is sufficiently large in later episodes, we have $\kappa_F \to 0$ and thus $\alpha \to 1$. We theoretically derive an upper bound $\kappa_F \leq \frac{\kappa_S}{\lambda\beta+1}$ (where $\kappa_S$ is the curvature of $F_{div}(\cdot)$), which goes to zero when $\lambda \to \infty$ (see Appendix A).

## 4   Experiments

We evaluate CHER and compare to state-of-the-art baselines on several challenging robotic manipulation tasks in simulated MuJoCo environments [Todorov et al., 2012]. In particular, we will use a simple Fetch environment as a toy example and Shadow Dexterous Hand environments from OpenAI Gym [Brockman et al., 2016]. It is worth noting that the Shadow Dexterous Hand environments are also the most difficult environments amongst OpenAI's robotics environments.

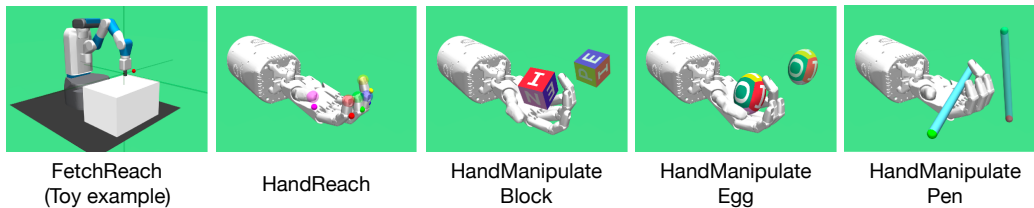

| FetchReach (Toy example) | HandReach | HandManipulate Block | HandManipulate Egg | HandManipulate Pen |

Figure 1: The Fetch and four Shadow Dexterous Hand environments.

## 4.1 Environments

In Figure 1, there are FetchReach environment and four Shadow Dexterous Hand environments: HandReach, Block manipulation (HandManipulateBlockRotateXYZ-v0), Egg manipulation (HandManipulateEggFull-v0) and Pen manipulation (HandManipulatePenRotate-v0). The FetchReach environment is based on the 7-DoF (degrees of freedom) Fetch robotics arm with a two-fingered parallel gripper. Each action $a_t$ is a 3-dimensional vector specifying the desired gripper movement in Cartesian coordinates and the gripper keeps closing during the process of reaching some target location. Each observation is the state of the robot. In the simulated environments, the shadow Dexterous Hand is an anthropomorphic robotic hand with 24 DoF, in which 20 joints can be controlled independently whereas the remaining ones are coupled joints. In all the four hand environments, each action $a_t$ is a 20-dimensional vector containing the absolute position control for all non-coupled joints of the hand. Each observation includes the 24 positions and the associated velocities of the 24 joints. To represent an object that is manipulated, the environment provides the object's Cartesian position and rotation represented by a 7-dimensional vector, as well as its linear and angular velocities. The rewards are sparse and binary: the agent receives a reward of 0 if the goal has been achieved (within some task-specific tolerance) and $-1$ otherwise.

In FetchReach, the goal of reaching task is a 3-dimensional vector describing the target position of an object (or the end-effector for reaching) and the achieved goal is the position of the gripper. We use Euclidean distance for $\text{dis}(g_i, g_j)$. In HandReach, the goal of reaching task is a target position and the desired goal is the position of fingertips. In Block and Pen manipulations, the goal of manipulation tasks is the rotation of a target pose and the achieved goal is the rotation of the block/pen. In Egg manipulation, the goal of manipulation task is the rotation and location of a target pose and the achieved goal is the rotation and location of the egg.

## 4.2 Baselines

Our evaluation of different methods is based on DDPG. We use different methods to select/sample hindsight experiences to replay and train policies on the environments issuing sparse rewards. We compare CHER with the following baselines:

- DDPG [Lillicrap et al., 2015], a model-free RL algorithm for continuous control. It learns a deterministic policy by a stochastic counterpart to explore during training.
- DDPG+HER [Andrychowicz et al., 2017], which samples hindsight experiences uniformly for replay.
- DDPG+HEREBP [Zhao and Tresp, 2018], which uses an energy function to evaluate trajectories and prioritize hindsight experiences with large energy.

The comparison between dense and sparse rewards has been presented in Plappert et al. [2018] and it has shown the advantage of using sparse rewards.

## 4.3 Training Setting

For all environments except FetchReach, we train policies on a single machine with 20 CPU cores. Each core generates experiences by using two parallel rollouts with MPI for synchronization. We train each agent for 50 epochs with batch size 64. Hyperparameters are nearly the same as in Andrychowicz et al. [2017]. In CHER, we use $|B| = 128$, $|A| = k = 64$ and $|b| = m = 3$ for Algorithm 1. We evaluate the policies after each epoch by performing 10 deterministic test rollouts per MPI worker, and then compute the test success rate by averaging across rollouts and MPI workers. In all cases, we

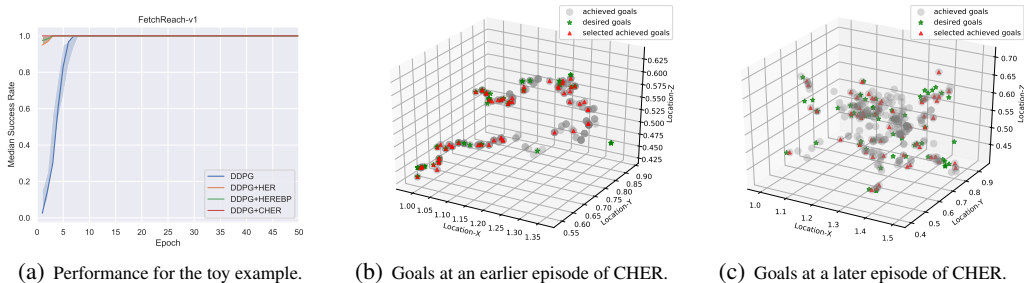

(a) Performance for the toy example.  (b) Goals at an earlier episode of CHER.  (c) Goals at a later episode of CHER.

Figure 2: Toy example – FetchReach. (a) CHER learns much faster than other RL methods. (b) The red points (selected achieved goals) compose a diverse and representative subset of the gray points (all achieved goals), but some are not close to any green point (desired goals) since CHER prefers diversity than proximity in earlier episodes. (c) Most red points are close to some green points due to the large proximity in later episodes' selection criteria, but some regions with many gray points concentrated do not contain any red point since CHER prefers proximity more than diversity.

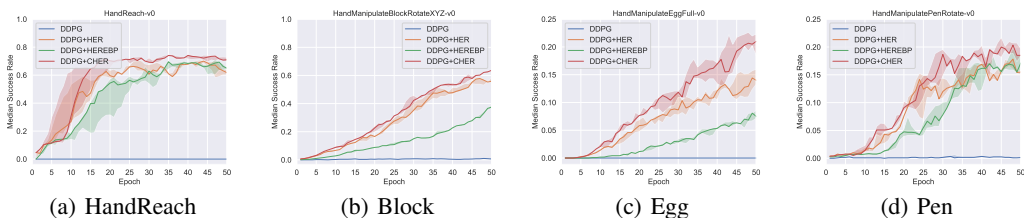

(a) HandReach  (b) Block  (c) Egg  (d) Pen

Figure 3: Performance for all four hand environments (Block: $\lambda_0 = 0$; others: $\lambda_0 = 1$).

repeat each experiment with 5 different random seeds and report their performance by computing the median test success rate as well as the interquartile range.

## 4.4 Toy Example

To quickly prove the concept of our idea, we first study it in a simple environment, FetchReach, as a toy example. We train policies by using one CPU core.

Figure 2(a) depicts the median test success rate for the FetchReach environment. FetchReach is known as a very simple environment and can be easily learned by our approach. The results show that DDPG+CHER learns faster than all other baselines. Vanilla DDPG can also reach $100\%$ success rate at last but much later than DDPG+CHER. DDPG+HEREBP performs similarly to DDPG+HER on this simple task.

Figure 2(b)(c) visualize the selected desired goals $g$ (green stars), all the achieved goals $B$ (grey circles), and the achieved goals $A \subseteq B$ selected by Algorithm 1 (red triangles) at an earlier episode (left) and a later episode (right) of DDPG+CHER. In the earlier episode, the achieved goals selected into $A$ averagely spread on the manifold of all the achieved goals $B$, implying that $A$ is a diverse and representative subset of $B$. There are regions that contain several selected goals far away from any desired goal, since the proximity plays a minor role in earlier episodes while the diversity dominates the selection criteria. In the later episode, in contrast, most of the achieved goals selected into $A$ gather around some desired goals, and there are regions where many unselected goals gather but none is selected, which indicates that the proximity dominates over the diversity in the selection.

## 4.5 Benchmark Results

Figure 3 reports how the median test success rate achieved by all methods improves during learning in the four hand environments. Similar to what is shown in the FetchReach environment, DDPG+CHER significantly outperforms the other baselines. The results also show that DDPG can easily fail in these environments, but DDPG+HER is able to learn partly successful policies in all environments. Surprisingly, DDPG+CHER has got significant improvement on Egg and Pen manipulation tasks, as

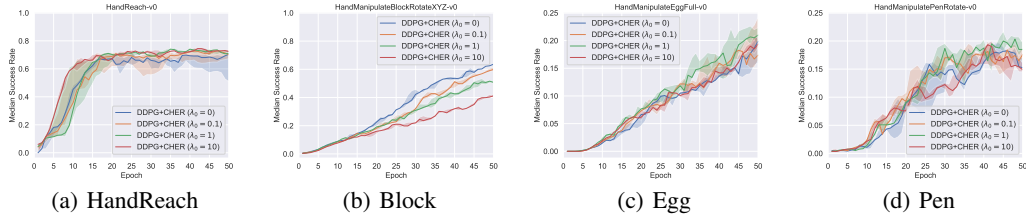

| (a) HandReach | (b) Block | (c) Egg | (d) Pen |

Figure 4: Performance of DDPG+CHER with different initial $\lambda_0$ for all four hand environments.

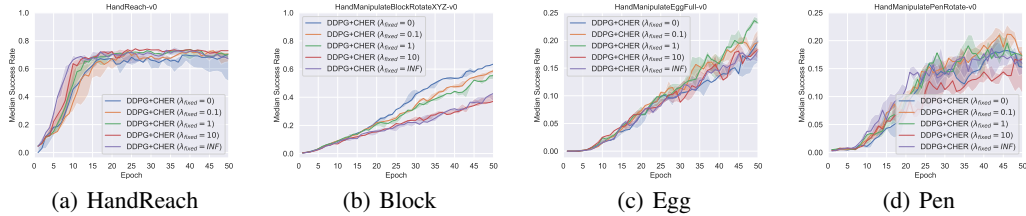

| (a) HandReach | (b) Block | (c) Egg | (d) Pen |

Figure 5: Ablation study of DDPG+CHER with $\lambda$ fixed ($\lambda_{fixed}$) for all four hand environments.

shown in Figures 3(c) and 3(d). These tasks are known to be difficult because the objects often drop down. With curriculum learning equipped in CHER, the agent quickly learns the way of holding the object. In summary, DDPG+CHER with curriculum learning that selects hindsight experiences for replay effectively improves the performance of DDPG+HER.

Figure 4 reports the success rate of DDPG+CHER using different initialization $\lambda_0$ for $\lambda$. It shows that promoting different amount of proximity in selection affects the performance. When $\lambda_0 = 0$, i.e., starting without any proximity preferred, the performance degrades. It also shows that too large proximity does not improve the performance.

In Figure 5, we test the performance of DDPG+CHER with $\lambda$ fixed at different values, where $\lambda$ =INF refers to proximity-only. Compared to DDPG+CHER using a curriculum of increasing $\lambda$ in Figure 3, the performance of CHER can significantly vary when using different $\lambda_{fixed}$, and some can perform much worse. In contrast, DDPG+CHER with a gradually increasing $\lambda$ usually results in a smoother and more stable learning process that can rapidly learn to accomplish challenging tasks.

## 5   Conclusion

The main contributions of this paper are summarized as follows: (1) We introduce "Goal-and-Curiosity-driven Curriculum Learning" for Hindsight Experience Replay (HER). To our knowledge, the resulted Curriculum-guided HER (CHER) is the first work that adaptively selects failed experiences for replay according to their compatibility and usefulness to different learning stages of deep RL; (2) We show that a large diversity is beneficial to earlier exploration, while a large proximity to the desired goals is essential for effective exploitation in later stages; (3) We show that the sample efficiency and learning speed of off-policy RL algorithms can be significantly improved by CHER. We attribute this to the global knowledge learning on a set of failed experiences, which breaks the constraint of local one-episode experience towards more robust strategies; (4) We apply CHER to several challenging continuous robotics environments with sparse rewards, and demonstrate its effectiveness and advantage over other HER-based approaches; (5) CHER does not make assumptions on tasks and environments, and can potentially be generalized to other more complicated tasks, environments and settings.

### Acknowledgments

We would like to thank Tencent AI Lab and Robotics X for providing an excellent research environment that made this work possible. Also, we would like to thank the anonymous reviewers.

## Footnotes

*Correspondence to: Meng Fang <mfang@tencent.com> and Tianyi Zhou <tianyizh@uw.edu>.

[1]Our code is available at `https://github.com/mengf1/CHER`.

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
