[Supplementary Material]

# Appendix A

**Curvature dependent approximation bound of Submodular maximization in Eq. (3)**

The combinatorial optimization problem in Eq. (3) is a submodular maximization with monotone non-decreasing submodular objective and a rank constraint. Simple vanilla greedy algorithm can yield an approximation factor of $\alpha = 1 - e^{-1}$. This result can be further improved when the objective $F(A)$ is close to modular, e.g., when $\lambda$ is sufficiently large. We therefore have the following Lemma:

**Lemma 1.** *Let $F(A) = \lambda F_M(A) + F_S(A)$ where $F_S(\cdot)$ is a monotone non-decreasing submodular function with curvature $\kappa_S$, $F_M(\cdot)$ is a non-negative modular function, both defined on ground set $V$, and $\lambda \geq 0$. Then $\kappa_F \leq \frac{\kappa_S}{\lambda\beta+1}$ where $\beta = \min_{j \in V} F_M(j)/F_S(j)$.*

*Proof.* We have

$$
\begin{aligned}
\kappa_F &= 1 - \min_{j \in V} \frac{\lambda F_M(j) + F_S(j|V \setminus j)}{\lambda F_M(j) + F_S(j)} \\
&= \max_{j \in V} \frac{F_S(j) - F_S(j|V \setminus j)}{\lambda F_M(j) + F_S(j)} \\
&= \max_{j \in V} \frac{1 - \frac{F_S(j|V \setminus j)}{F_S(j)}}{\frac{\lambda F_M(j)}{F_S(j)} + 1} \\
&\leq \frac{\kappa_S}{\min_{j \in V} \frac{\lambda F_M(j)}{F_S(j)} + 1} = \frac{\kappa_S}{\lambda\beta + 1}
\end{aligned}
$$

where $\beta \triangleq \min_{j \in V} \frac{F_M(j)}{F_S(j)}$. $\qquad\square$