[Reviews · NeurIPS 2019]

Reviewer 1



The paper borrows tools from combinatorial optimization (i.e. for the facility location problem) in order to select hindsight goals that simultaneously has high diversity and also being close to the desired goals. As mentioned, the similarity metric used for the proximity term seems to require domain knowledge that euclidean distance works well for this task. This may be problematic if we have obstacles that mislead the euclidean distance, or in another environment where it is less obvious what the similarity metric can be. I am aware that this dense similarity metric is only used for hindsight goals, and that the underlying Q function/policy is still trained on the sparse reward (without the bias). There are several related works that can be discussed and potentially benchmarked against in terms of hindsight goal sampling schemes: Sampling from ground truth distribution half the time for relabeling, and using future the other time (in Appendix). A. Nair, et. al. Visual Reinforcement Learning with Imagined Goals. NIPS 2018. A heuristic goal sampling scheme: D. Warde-Farley, et. al. Unsupervised Control Through Non-Parametric Discriminative Rewards. ICLR 2019 The paper supported its claims on the Goal and Curiosity Driven Curriculum (GCC) learning with qualitative plots of the selected hindsight goals over the course of training. The plots seems to indicate that indeed the earlier episode hindsight goals have higher diversity while latter episodes are closer to the desired goals. The ablation studies on the lambda_0 value indicates that having both the diversity and proximity terms can affect the performance. To prove that lambda curriculum is necessary, I think that it will also be helpful to compare different *fixed* value of lambda (i.e. no curriculum) vs with the lambda curriculum. The paper is fairly clearly written, with understandable high level ideas. There are some clarification details/suggestions: What is the similarity metric used, Equation 1 or 2 (or neither) for the experiments? What is the \eta and \lambda_0 value? This will also tell us how large \lambda gets by the time by the end of training. How sensitive is the performance to this parameter? Figure 4 gives a nice qualitative view of the selected achieved goals in relation to the desired goals and achieved goals. Having a quantitative value can also be valuable, i.e. plotting the value of F_prox(A), F_div(A), over the course of training. The large performance in the hand manipulate pen rotate task deserves some attention, as previous approaches so far have not been able to make much improvement. While the method seems more of a heuristic, I think that the approach proposed will benefit the goal-conditioned RL community. *** Post Author Rebuttal Comments *** Thank you to the authors for their response. I am fairly satisfied with the author response: - Given the additional ablations in their rebuttal that I have specifically asked for (i.e. with the fixed \lambda, sensitivity on the \eta parameters, F_div/F_prox curves), they have proved the importance of having the curriculum to balance the proximity/diversity (exploit vs explore). - On using Euclidean distance proximity, the authors also reasonably addressed this in their rebuttal, especially emphasizing that the main contributing point is on balancing between the proximity/diversity, *given* a metric. I will still encourage them to have a discussion about having the right metric for the domain in their final draft. There is unfortunately probably not enough space in the publication format to do a thorough investigation with other metrics / domain spaces other than L2, so I am ok to leave that to future work. - On related works: as they pointed out the works I mentioned are not directly comparable (i.e. using image input instead of object position state representations), but those works also touch on balancing some sense of diversity in the hindsight goals (i.e. via sampling from prior) versus the using seen states. So I did not expect them to try to directly compare by applying CHER to domains with image inputs. Overall I increased my score from my original review. *************

Reviewer 2



Summary: The paper is based on the observation that experience sampling in HER is inefficient because certain pseudo-goals are irrelevant to the actual goal. It proposes to train a subset of experiences that maximizes both proximity to the actual goal and diversity/representativeness (described by the facility location function). In addition, it scales up the proximity coefficient exponentially throughout training, so that HER converges more quickly. The final algorithm, termed CHER, outperforms the baseline DDPG significantly on 2 out of 4 hand-manipulation tasks.   Strengths: 1. Novel idea on balancing exploitation (near-actual-goal sampling) and exploration (diverse goal sampling) for HER. 2. CHER clearly attains better performance than HER on the proposed tasks. 3. The paper is well-written and easy to follow. Weaknesses: 1. There is no discussion on the choice of "proximity" and the nature of the task. On the proposed tasks, proximity on the fingertip Cartesian positions is strongly correlated with proximity in the solution space. However, this relationship doesn't hold for certain tasks. For example, in a complicated maze, two nearby positions in the Euclidean metric can be very far in the actual path. For robotic tasks with various obstacles and collisions, similar results apply. The paper would be better if it analyzes what tasks have reasonable proximity metrics, and demostrate failure on those that don't. 2.  Some ablation study is missing, which could cause confusion and extra experimentation for practitioners. For example, the \sigma in the RBF kernel seems to play a crucial role, but no analysis is given on it. Figure 4 analyzes how changing \lambda changes the performance, but it would be nice to see how \eta and \tau in equation (7) affect performance. Minor comments: 1. The diversity term, defined as the facility location function, is undirected and history-invariant. Thus it shouldn't be called "curiosity", since curiosity only works on novel experiences. Please use a different name. 2. The curves in Figure 3 (a) are suspiciously cut at Epoch = 50, after which the baseline methods seem to catch up and perhaps surpass CHER. Perhaps this should be explained.

Reviewer 3



The empirical results were difficult to interpret. Some of this difficulty was due to ambiguity of the results themselves, but the rest could have been addressed in the discussion. The lack of consistency in results across tasks, coupled with a weak discussion of these experiments are concerning. However, that the lambda=1 setting consistently outperforms the baselines and ablations suggests that the general technique is worth trying as a drop-in replacement for uniform sampling in experience-replay UVFA settings. Analysis includes a proof on the bound of the sub optimality of the sampling strategy (which is nice) but it would have been helpful to include an empirical evaluation what effect sub optimality in sampling actually has on agent performance. Is performance highly dependent on getting sampling just right, or is anything that makes sampling more greedy sufficient? Relatedly, it would have been interesting to compare with simpler prioritized-replay mechanisms, e.g. directly using the goal-similarity metric in a priority queue, particularly since this is easy to implement. "A diverse subset A of achieved goals encourage the agent to explore new and unseen areas of the environment and learn to reach 146 different goals." > In a HER/UVFA setting it seems that the online choice of goal is the biggest factor in determining exploration, vs. what is backed up offline. I'd expect that in some cases sampling diverse goals could actually decrease exploration for a given goal by removing delusions in the value function. Overall I think the connection between prioritized sampling, which this paper focuses on, and the exploration-and-exploitation trade-off, which is typically viewed as an online choice or a reward augmentation, is tenuous and warrants further discussion. In Fig 3) CHER offers no benefit over HER and HEREBP baselines for tasks b & c, but is significant on tasks a & d. To what do the authors attribute this difference? I also find it suspicious that the HER and HEREBP traces are nearly identical for all experiments, and even have similar dips in what is typically noise. In Fig 4) tasks c & d, which are the harder and more interesting tasks, the effect of lambda is quite small, which suggests that the benefit for fig 3d vs DDPG-HER is mainly an effect of vs , and not the balance of greedy or diverse sampling. Might this be explained by other parameters or tuning for the baseline implementations? This result seems particularly surprising since in fig3a, which is a similar task, CHER and DDPG-HER had equivalent performance. To what can we attribute this inconsistency? On tasks a & b lambda 100 > 0 and 10 > 0.1, suggesting that 1 is optimal, but that being too greedy is better than being to diverse. This is an interesting result, but doesn’t seem to hold in c&d. Plot titles and axis legends difficult to read. What is the compute cost of the proposed method? Running an iterative optimization inside the batch-sampling step of a Deep-Rl algo sounds expensive.

[Author Response · NeurIPS 2019]

We appreciate the reviewers' efforts and suggestions (in blue)! We will address them all in the next version, and
cite/discuss all the papers mentioned by reviewers. We will answer the shared question and then reply to each reviewer.
*Shared*: ● (Improvements) How to choose similarity metric (or proximity) for different tasks? 1) We focus on multi-goal ML tasks
that HER and many works aim to address, on which CHER using proximity in Eq.1 with Euclidean distance can achieve
compelling results. Although Euclidean distance is not universal for arbitrary tasks, it works generally well over many
tasks important to RL community. 2) Every RL system more or less relies on domain knowledge such as some physical
laws. Tasks may prefer different distance metrics, but most physical systems have their own predefined ones, e.g., for
environment with obstacles or a maze, it is natural to have proximity as the length of the shortest legal path to the goal;
for a smooth surface, geodesic distance is a better choice. 3) Our key idea, a curriculum with increasing proximity and
decreasing diversity, is not limited to Euclidean distance and can be applied with any predefined proximity.
*Reviewer 1*: ● A theoretical motivation for having the curriculum on the $\lambda$ value in the first place was not given. A critical theoretical
motivation of having $\lambda$ is to balance exploration-exploitation trade-off as in online learning problems.
● (Improvements) Comment on some of the related work on hindsight goal sampling. We will add discussion of those works. They
are not directly comparable due to different task settings: their observations are raw pixels, while HER and CHER use
physical positions. For fair comparison, we need to modify either HER/CHER or baselines, like [Nair et. al, 2018]
modifying HER to minimize pixel-MSE. But this makes the original task harder with extra cost of training a VAE
handling raw pixels. [Warde-Farley et.al, 2019] are different to CHER in: 1) it learns a reward function to address the
sparse reward problem, and 2) it samples the goal buffer as uniform as possible.
● (Improvements) Ablations on fixed $\lambda$ values. Plotting the value of $F_{prox}(A)$, $F_{div}(A)$ over the course of training. The left 4 plots
below report the performance of 6 different fixed $\lambda$ values on the four tasks, where $\lambda =$INF refers to proximity(goal-
similarity)-only. Compared to Fig.3-4 using $\lambda$-curriculum (Fig.4 shows CHER initialized with different $\lambda_0$), fixed $\lambda$
performs much worse. The $6^{th}$ plot below shows how $F_{prox}(A)$ and $F_{div}(A)$ change during training.

● What is the $\eta$ and $\lambda_0$ value? How sensitive is the performance to this parameter? Fig.4 shows that CHER's performance remains
stably good for $\lambda_0 \in [0.1, 10]$. The $5^{th}$ plot above (using $\lambda_0 = 1$) shows that CHER is also robust to different $\eta$. They
indicates that CHER is robust to the choices of $\eta$ and $\lambda_0$. We use $\eta = 0.0001$ and $\lambda_0 = 1$ for all the tasks in experiments.
*Reviewer 2*: ● (Improvements) Ablation study of different $\sigma$ in the RBF kernel. How $\eta$ and $\tau$ in Eq.7 affect performance? We adopt
an adaptive $\sigma$ widely used in kernel methods: the average distance over all $(x, y)$ pairs. It practically has promising
performance while saves tuning cost. Result of different $\eta$ is shown in the $5^{th}$ plot above. $\tau$ is the episode number.
● (Improvements) The diversity term shouldn't be called "curiosity" We will change it to "diversity".
● (Improvements) The curves in Fig.3(a) are suspiciously cut at Epoch=50, after which the baseline methods seem to catch up and perhaps
surpass CHER. They saturate and won't surpass CHER later. Zooming in Fig.3(a) at Epoch=50 also shows that the orange
curve (our method) increases the fastest, while the baselines are either decreasing or increasing slower.
*Reviewer 3*: ● (Improvements) An empirical evaluation what effect sub optimality in sampling actually has on agent performance. In
the rightest plot above, we report the time costs and success rates for StochasticGreedy with different sub-sampling
sizes. It shows that small sub-sampling size improves efficiency but does not harm the optimality.
● (Improvements) Compare with simpler prioritized-replay mechanisms, e.g. directly using the goal-similarity metric in a priority queue.
In above 4 plots of fixed $\lambda$, the success rate of solely using goal-similarity/proximity($\lambda =$INF) is lower than smaller
fixed-$\lambda$ and $\lambda$-curriculum in Fig.3-4. This indicates the importance of diversity for HER.
● In Fig.3, CHER offers no benefit over HER and HEREBP baselines for tasks b & c, but is significant on tasks a & d. To what do the authors
attribute this difference? Task b, c and d are all rotating tasks. Comparing to d, b & c have shapes (block and egg) easier
to handle, and the proximity and diversity of different achieved goals are more similar to each other since the shapes are
more rotate-invariant. Hence, CHER makes less difference on b & c. Nevertheless, it still achieves the best performance
on b & c and significantly outperforms the best baseline by $4\%$ and $1.5\%$ (note the baseline already has $> 95\%$).
● (Improvements) Inconsistency of c & d between Fig.3 and Fig.4: the effect of $\lambda$ is quite small in Fig.4 for both c & d. In Fig.4, on tasks
a & b, $\lambda = 1$ is optimal, but doesn't seem to hold in c & d. Sorry for the typo in Fig.4's caption: it shows CHER initialized with
different $\lambda_0$. It shows that the effect of changing $\lambda_0$ in $[0.1, 10]$ is small, and $\lambda_0 \simeq 1$ performs the best, same as Fig.4(a)-
(b). It actually exhibits the robustness of CHER to $\lambda_0$: its remarkable performance is mainly resulted from the dynamic
curriculum rather than careful tuning of $\lambda_0$. We tuned all the baselines for their best performance, which are consistent
with previous papers about HER on the same tasks. In Fig.3(a), CHER reaches a success rate of $78\%$ only after 10 epochs
while DDPG-HER spent 50 epochs. CHER is much more efficient for its careful selection of curriculum. Although
a & d are similar tasks, their robots have different mechanical structures, which results in different performance.
● Plot titles and axis legends difficult to read. We will try our best to improve their readability.
● What is the compute cost of the proposed method? Running an iterative optimization inside the batch-sampling step of a Deep-RL
algorithm sounds expensive. The only extra computation of CHER (comparing to HER) is to run StochasticGreedy in
Line 12 of Algorithm 2, which only needs $mk$ evaluations of Eq.5 (note the similarities in Eq.5 are invariant and
pre-computed). In our experiments, $mk = 192$ and the extra computation is ignorable ($< 5\%$ of the total training time).

[Meta-Review · NeurIPS 2019]

The paper proposes a method that improves over the Hindsight Experience Replay (HER) method by prioritizing training experiences whose pseudo-goals are closer to the actual goals. Goals are sampled according to a score that balances between (1) proximity to desired goals and (2) diversity of achieved goals chosen. The paper is well-written, the proposed method is new and interesting. The experiments on simulated robotic manipulation tasks also support the claims for the paper.